

# A line integral-based method to partition climate and catchment effects on runoff

Mingguo Zheng[1, 2*]

[1] Guangdong Key Laboratory of Agricultural Environment Pollution Integrated Control, Guangdong Institute of Eco-environment Science & Technology, Guangzhou 510650, China
[2] Key Laboratory of Water Cycle and Related Land Surface Processes, Institute of Geographic Sciences & Natural Resources Research, Chinese Academic of Sciences, Beijing 100101, China
Correspondence: Mingguo Zheng (mgzheng@soil.gd.cn)

## Abstract

It is a common task to partition synergistic impacts of a number of drivers in the environmental sciences. However, there is no mathematically precise solution to the partition. Here I presented a line integral-based method, which concerns about the sensitivity to the drivers throughout their evolutionary path so as to ensure a precise partition. The method reveals that the partition depends on both the change magnitude and pathway (timing of change), and not on the magnitude alone unless for a linear system. To illustrate the method, I used the Budyko framework to partition the effects on the temporal change in runoff of climatic and catchment conditions for 21 catchments from Australia and China. The method reduced to the decomposition method when assumed a path along which climate change occurs first followed by an abrupt change in catchment properties. The method re-defines the widely-used concept of sensitivity at a point as the path-averaged sensitivity. The total differential and the complementary methods simply concern about the sensitivity at the initial or/and the terminal state, so that they cannot give precise results. The path-average sensitivity of water yield to climate conditions was found to be stable over time. Space-wise, moreover, it can be readily predicted even in the absence of streamflow observations, whereby facilitates evaluation of future climate effects on streamflow. As a mathematically accurate solution, the method provides a generic tool to conduct the quantitative attribution analyses.

**Keywords:** Runoff; Climate change; Human activities; Attribution analysis; Budyko

## 1 Introduction

It is often needed to quantify the relative roles of a few drivers to the observed changes of interest in the environmental sciences. In the hydrology community, diagnosing the relative contributions of climate change and human activities to runoff is of great relevance to the researchers and managers as both climate change and human activities have pose global-scale impact on hydrologic cycle and water resources (Barnett *et al.*, 2008; Xu *et al*., 2014; Wang and Hejazi, 2001). To date, unfortunately, the quantitative attribution analysis of the runoff changes remains a challenge (Wang and Hejazi, 2001; Berghuijs and Woods, 2016; Zhang *et al.*, 2016); this is to a considerable degree due to a





lack of a mathematically precise method to decouple synergistic and often confounding impacts of
climate change and human activities.
Numerous studies have detected the long term variability in runoff and attempted to partition the
effects of climate change and human activities by means of various methods (Dey and Mishra, 2017).
Among them are the paired-catchments method and the hydrological modeling method. The paired-
catchment method is believed to be able to filter the effect of climatic variability and thus isolate the
runoff change induced by vegetation changes (Brown *et al.*, 2005). However, the method is
capital intensive. Particularly, it generally involves small catchments and is challenged when
extrapolating to large catchments (Zhang *et al.*, 2011). The physical-based hydrological models often
suffer from limitations including high data requirement, labor-intensive calibration and validation
processes, and inherent uncertainty and interdependence in parameter estimations (Binley *et al.*, 1991;
Wang *et al.*, 2013; Liang *et al.*, 2015). Interest then turns to the conceptual models over recent years,
such as the Budyko-type equations (see Section 2.1).
Within the Budyko framework, a large number of studies (Roderick and Farquhar, 2011; Zhang
*et al.*, 2016) have used the total differential as a proxy for the runoff change and further evaluated
hydrological responses to climate change and human activities (hereafter called the total differential
method). The total differential, however, is essentially a first-order approximation of the observed
change. It has been shown that the approximation has caused an error of the climate impact on runoff
ranging from 0 to 20 mm (or -118 to 174%) over China (Yang *et al.*, 2014). The total differential
method directly used the partial derivatives of runoff to estimate the sensitivities of runoff to climate
and catchment conditions. Most studies applied the forward approximation of the runoff change, *i.e.*,
using the sensitivities at the initial state while calculation (e.g. Roderick and Farquhar, 2011). The
elasticity method proposed by Schaake (1990) is also based on the total differential expression
(Sankarasubramanian *et al.*, 2001; Zheng *et al.*, 2009). The method uses the "elasticity" concept to
assess the climate sensitivity of runoff. The elasticity coefficients, however, have been estimated in an
empirical way and is not physically sound (Roderick and Farquhar, 2011; Liang *et al.*, 2015).
The so-called decomposition method developed by Wang and Hejazi (2011) has also been
widely used. The method assumes that climate changes drive a shift along a Budyko curve and then
human interferences cause a vertical shift from the Budyko curve to another. Under this assumption, the
method directly extrapolates the Budyko models calibrated using observations of the reference period,
in which human impacts remain minimal, to determine the human-induced changes in runoff occurred
during the evaluation period.
Recently, Zhou *et al.* (2016) established a Budyko complementary relationship for runoff and
applied it to partitioning the climate and catchment effects. Superior to the total differential method, the
method culminates with yielding a no-residual partition. Nevertheless, the method depends on a given
weighted factor, which is determined in an empirical but not a precise way. Furthermore, Zhou *et al.*
(2016) argued that the partition is not unique in the Budyko framework as the path of the climate and
catchment changes cannot be uniquely identified.
Actually, a precise partition remains difficult even given a  a precise mathematical model. This
can be illustrated by using a precise hydrology model $R = \mathrm{f}(x, y)$, where $R$ represents runoff, and $x$ and $y$
climate factors and catchment characteristics respectively. We assumed that $R$ changes by $\Delta R$ when $x$
changes by $\Delta x$ and $y$ by $\Delta y$, *i.e.* $\Delta R = f(x + \Delta x, y + \Delta y) - f(x, \mathrm{y})$. To determine the effect of $x$ on $\Delta R$,



*i.e.* $\Delta R_x$, a common practice is to assume that $y$ remains constant when $x$ changes by $\Delta x$. We thus get:
$\Delta R_x = f(x + \Delta x, y) - f(x, y)$. Similarly, we can get: $\Delta R_y = f(x, y + \Delta y) - f(x, y)$. Although the
derivation seems quite reasonable, it is problematic as the sum of $\Delta R_x$ and $\Delta R_y$ is not equal to $\Delta R$.
Further examination shows that a variable's effect on $R$ should differ depending on the changing path.
For example, $\Delta R_x = f(x + \Delta x, y) - f(x, y)$ and $\Delta R_y = f(x + \Delta x, y + \Delta y) - f(x + \Delta x, y)$ if $x$ changes first
and $y$ subsequently (Note that the sum of $\Delta R_x$ and $\Delta R_y$ equals $\Delta R$ now). If $y$ changes first and $x$
subsequently, in contrast, the expressions become: $\Delta R_x = f(x + \Delta x, y + \Delta y) - f(x, y + \Delta y)$ and
$\Delta R_y = f(x, y + \Delta y) - f(x, y)$. In case of $x$ and $y$ changing simultaneously, unfortunately,
current literature seems not to provide a mathematically precise solution.
The aims of this work are to propose a new and mathematically precise method to conduct
quantitative attribution to the drivers. The method is based on the line integer (called the LI method
hereafter) and takes account of the sensitivity throughout the evolutionary path of the drivers, thus
revising the widely-used concept of sensitivity at a point as the path-averaged sensitivity. To present
and evaluate the method, I decomposed the relative influences of climate and catchment conditions on
runoff within the Budyko framework using data from 21 catchments from Australia and China. I also
examined the spatio-temporal variability of the path-averaged sensitivities of runoff to climatic and
catchment conditions and assessed their spatio-temporal predictability.

## 2 Methodology

### 2.1 The Budyko Framework and the MCY equation

Budyko (1974) argued that the mean annual evapotranspiration ($E$) is largely determined by
water and energy balance of a catchment. Using precipitation ($P$) and potential evapotranspiration ($E_0$)
as proxies for water and energy availabilities respectively, the Budyko framework
relates evapotranspiration losses to the aridity index defined as the ratio of $E_0$ over $P$. The Budyko
framework has gained wide acceptance in the hydrology community (Berghuijs and Woods, 2016;
Sposito, 2017). Over past decades, a number of equations have been developed to describe the
framework. Among them, the Mezentsev-Choudhury-Yang's equation (Mezentsev, 1955; Choudhury,
1999; Yang *et al.*, 2008) (Called the MCY equation hereafter) has been widely accepted and was used
here:

$$\frac{E}{P} = \frac{E_0/P}{(1 + (E_0/P)^n)^{1/n}} \tag{1}$$

where $n \in (0, \infty)$ is an integration constant that is dimensionless, and represents catchment properties.
Eq. (3) requires a relative long time scale whereby the water storage of a catchment is negligible and the
water balance equation reduces to be $R = P - E$, where $R$ denotes mean annual runoff. Here I adopted a
"tuned" $n$ value that can get exact agreement between the calculated $E$ by Eq. (1) and that actually
encountered ($= P - R$).
The partial differentials of $R$ with respect to $P$, $E_0$, and $n$ are given as:





$$\frac{\partial R}{\partial P} = R_P(P, E_0, n) = 1 - \frac{E_0^{n+1}}{(P^n + E_0^n)^{1/n}}$$ (2a)

$$\frac{\partial R}{\partial E_0} = R_{E_0}(P, E_0, n) = -\frac{P^{n+1}}{(P^n + E_0^n)^{1/n}}$$ (2b)

$$\frac{\partial R}{\partial n} = R_n(P, E_0, n) = \frac{-E_0 P n^{-1}}{(P^n + E_0^n)^{1/n}} \left[ \frac{\ln(P^n + E_0^n)}{n} - \frac{P^n \ln P + E_0^n \ln E_0}{P^n + E_0^n} \right]$$ (2c)

## 2.2 The theory of the line integral-based method
To present the LI method, we start by considering an example of a two-variable function $z = f(x,$
$y)$, which has continuous partial derivatives $\partial z / \partial x = f_x(x, y)$ and $\partial z / \partial y = f_y(x, y)$. Suppose that $x$ and $y$
varies along a smooth curve $L$ (e.g. $AC$ in Fig. 1) from the initial state $(x_0, y_0)$ to the terminal state $(x_N,$
$y_N)$, and $z$ co-varies from $z_0$ to $z_N$. Let $\Delta z = z_N - z_0$, $\Delta x = x_N - x_0$, and $\Delta y = y_N - y_0$. Our goal is to seek
for a mathematical solution to quantify the effects of $\Delta x$ and $\Delta y$ on $\Delta z$, i.e. $\Delta z_x$ and $\Delta z_y$. $\Delta z_x$ and $\Delta z_y$
should be subject to the constraint $\Delta z_x + \Delta z_y = \Delta z$.
As shown in Fig. 1, points $M_1(x_1, y_1), \ldots, M_{N-1}(x_{N-1}, y_{N-1})$ partition $L$ into $N$ distinct segments. Let
$\Delta x_i = x_{i+1} - x_i$, $\Delta y_i = y_{i+1} - y_i$, and $\Delta z_i = z_{i+1} - z_i$. For each segment, $\Delta z_i$ can be approximated as the
total differential $dz_i$: $\Delta z_i \approx dz_i = f_x(x_i, y_i)\Delta x_i + f_y(x_i, y_i)\Delta y_i$. We then have:
$\Delta z = \sum_{i=1}^{N} \Delta z_i \approx \sum_{i=1}^{N} f_x(x_i, y_i)\Delta x_i + \sum_{i=1}^{N} f_y(x_i, y_i)\Delta y_i$. We thus obtain an approximation of $\Delta z_x$ and $\Delta z_y$:
$\Delta z_x \approx \sum_{i=1}^{N} f_x(x_i, y_i)\Delta x_i$ and $\Delta z_y \approx \sum_{i=1}^{N} f_y(x_i, y_i)\Delta y_i$. Define $\tau$ as the maximum length among the $N$ segments.
The smaller the value of $\tau$, the closer to $\Delta z_i$ the value of $dz_i$, and then the better the approximations are.
The approximations would become exact in the limit $\tau \to 0$. Taking the limit $\tau \to 0$ then turns sum into
integrals and gives a precise expression (it is an informal derivation and please see Appendix A for a
formal one): $\Delta z = \lim_{\tau \to 0} \sum_{i=1}^{N} f_x(x_i, y_i)\Delta x_i + \lim_{\tau \to 0} \sum_{i=1}^{N} f_y(x_i, y_i)\Delta y_i = \int_L f_x(x, y)dx + \int_L f_y(x, y)dy$, where
$\int_L f_x(x, y)dx = \lim_{\tau \to 0} \sum_{i=1}^{N} f_x(x_i, y_i)\Delta x_i$ and $\int_L f_y(x, y)dy = \lim_{\tau \to 0} \sum_{i=1}^{N} f_y(x_i, y_i)\Delta y_i$ denote the line integral of $f_x$ and $f_y$
along $L$ (termed integral path) with respect to $x$ and $y$, respectively. $\int_L f_x(x, y)dx$ and $\int_L f_y(x, y)dy$ exist
provided that $f_x$ and $f_y$ are continuous along $L$. We thus obtain a precise evaluation of $\Delta z_x$ and $\Delta z_y$:
$\Delta z_x = \int_L f_x(x, y)dx$ (3a)
$\Delta z_y = \int_L f_y(x, y)dy$. (3b)
Mathematically, the sum of $\Delta z_x$ and $\Delta z_y$ persistently equals $\Delta z$, independent of the curve $L$
(Appendix B). If $f(x, y)$ is linear, then $f_x$ and $f_y$ are constant. Define $C_x = f_x(x, y)$ and $C_y = f_y(x, y)$, we
have $\Delta z_x = C_x \Delta x$ and $\Delta z_y = C_y \Delta x$. $\Delta z_x$ and $\Delta z_y$ are thus independent of $L$. If $f(x, y)$ is non-linear, in contrast,





both $\Delta z_x$ and $\Delta z_y$ varies with $L$, as was exemplified in Appendix C. Hence, the initial and the terminal
states, together with the path connecting them, determines $\Delta z_x$ and $\Delta z_y$ unless $f(x, y)$ is linear.
The mathematical derivation above applies to a three-variable function as well. By doing the line
integrals for the MCY equation, we obtain the desired results:
$$\Delta R_P = \int_L \frac{\partial R}{\partial P} dP \qquad\qquad (4a)$$
$$\Delta R_{E_0} = \int_L \frac{\partial R}{\partial E_0} dE_0 \qquad\qquad (4b)$$
$$\Delta R_n = \int_L \frac{\partial R}{\partial n} dn \qquad\qquad (4c)$$
where $\Delta R_P$, $\Delta R_{E_0}$, and $\Delta R_n$ denotes the effects on runoff change of $P$, $E_0$, and $n$, respectively. The sum of
$\Delta R_P$ and $\Delta R_{E_0}$ represents the effect of climate change, and $\Delta R_n$ are often related to human activities
although it probably includes the effects of other factors, such as climate seasonality (Roderick and
Farquhar, 2011; Berghuijs and Woods, 2016). $L$ denotes a three-dimensional curve along which climate
and catchment changes have occurred. I approximated $L$ as a union of a series of line segments. $\Delta R_P$,
$\Delta R_{E_0}$, and $\Delta R_n$ were finally figured out by summing up the integrals along each of the line segments (see
Section 2.3).
2.3 Using the LI method to determine $\Delta R_P$, $\Delta R_{E_0}$, and $\Delta R_n$ within the Budyko Framework
1)   Determining $\Delta R_P$, $\Delta R_{E_0}$, and $\Delta R_n$ assuming a linear integral path
.Given two consecutive periods and assumed that the catchment state has evolved from $(P_1, E_{01},$
$n_1)$ to $(P_2, E_{02}, n_2)$ along a straight line $L$. Let $\Delta P = P_2 - P_1$, $\Delta E_0 = E_{02} - E_{01}$, and $\Delta n = n_2 - n_1$, then the
line $L$ is given by the parametric equations: $P = \Delta P t + P_1$, $E_0 = \Delta E_0 t + E_{01}$, $n = \Delta n t + n_1$, $t \in [0,1]$. Given
the equations, Eq. (2) becomes a one-variable function of $t$, i.e., $\partial R / \partial P = R_P(t)$, $\partial R / \partial E_0 = R_{E_0}(t)$, and
$\partial R / \partial n = R_n(t)$. Then, $\Delta R_P$, $\Delta R_{E_0}$, and $\Delta R_n$ can be evaluated as:
$$\Delta R_P = \int_L \frac{\partial R}{\partial P} dP = \int_0^1 R_P(t) d(\Delta P t + P_1) = \Delta P \int_0^1 R_P(t) dt \qquad (5a)$$
$$\Delta R_{E_0} = \int_L \frac{\partial R}{\partial E_0} dE_0 = \int_0^1 R_{E_0}(t) d(\Delta E_0 t + E_{01}) = \Delta E_0 \int_0^1 R_{E_0}(t) dt \qquad (5b)$$
$$\Delta R_n = \int_L \frac{\partial R}{\partial n} dn = \int_0^1 R_n(t) d(\Delta n t + n_1) = \Delta n \int_0^1 R_n(t) dt \qquad (5c)$$
Unfortunately, I cannot figure out the antiderivatives of $R_P(t)$, $R_{E_0}(t)$, and $R_n(t)$ and have to make
approximate calculations. I divided the $t \in [0,1]$ interval into 1000 subintervals of the same width,
thereby setting $dt$ identically equal to 0.001. I then calculated $R_P(t)dt$, $R_{E_0}(t)dt$, and $R_n(t)dt$ for each
subinterval. Let $t_i = 0.001i, i \in [0,999]$ and is integer-valued, $\Delta R_P$, $\Delta R_{E_0}$, and $\Delta R_n$ was approximated as:
$$\Delta R_P \approx 0.001 \Delta P \sum_{i=0}^{999} R_P(t_i) \qquad\qquad (6a)$$
$$\Delta R_{E_0} \approx 0.001 \Delta E_0 \sum_{i=0}^{999} R_{E_0}(t_i) \qquad\qquad (6b)$$



$$\Delta R_n \approx 0.001 \Delta n \sum_{i=0}^{999} R_n(t_i)$$
(6c)

2) Dividing the evaluation period into a number of subperiods
I first determine a change point and divide the whole observation period into the reference and
evaluation periods. To determine the integral path, the evaluation period is further divided into a
number of subperiods. The Budyko framework assumes a steady state condition of a catchment and
therefore requires no change in soil water storage. Over a time period of 5-10 years, it is reasonable to
assume that changes in soil water storage are sufficiently small (Zhang *et al.*, 2001). Here I divided the
evaluation period into a number of 7-year subperiods with the exception for the last one, which varied
from 7 to 13 years in length depending on the length of the evaluation period.
3)     Determining $\Delta R_P$, $\Delta R_{E_0}$, and $\Delta R_n$ by approximating the integral path as a series of line segments
For a short period, the integral path $L$ can be considered as linear, which implies a uniform
change over time. If the change is not uniform over a given long period, the integral path $L$ can be fitted
using a number of line segments. Given a reference period and an evaluation period comprising $N$
subperiods, I assumed that the catchment state evolved from $(P_0, E_{00}, n_0)$, …, $(P_i, E_{0i}, n_i)$, …, to $(P_N, E_{0N}, n_N)$, where the subscript "0" denotes the reference period, and "$i$" and "$N$" denotes the $i$th and the last
$n_N)$, where the subscript "0" denotes the reference period, and "$i$" and "$N$" denotes the $i$th and the last
subperiods of the evaluation period, respectively. I used a series of line segments $L_1, L_2, …, L_N$ to
approximate the integral path $L$, where the initial point of $L_{i+1}$ is the terminal point of $L_i$, and $L_i$ connects
points $(P_{i-1}, E_{0,i-1}, n_{i-1})$ with $(P_i, E_{0i}, n_i)$ and $L_1$ connects $(P_0, E_{00}, n_0)$ with$(P_1, E_{01}, n_1)$. Then $\Delta R_P$, $\Delta R_{E_0}$, and
$\Delta R_n$ are determined as the sum of the integrals along each of the line segments, which was calculated
using Eq. (6).
2.4 The total-differential, decomposition and complementary methods
To evaluate the LI method, I compared it with the decomposition method, the total differential
method, and the complementary method. The total differential method approximated $\Delta R$ as $dR$:
$$\Delta R \approx dR = \frac{\partial R}{\partial P}\Delta P + \frac{\partial R}{\partial E_0}\Delta E_0 + \frac{\partial R}{\partial n}\Delta n = \lambda_P \Delta P + \lambda_{E_0}\Delta E_0 + \lambda_n \Delta n$$
(7)

where $\lambda_P = \partial R/\partial P$, $\lambda_{E_0} = \partial R/\partial E_0$, and $\lambda_n = \partial R/\partial n$, representing the sensitivity coefficient of $R$ with
respect to $P$, $E_0$, and $n$, respectively. Within the total differential method, $\Delta R_P = \lambda_P \Delta P$, $\Delta R_{E_0} = \lambda_{E_0}\Delta E_0$, and
$\Delta R_n = \lambda_n \Delta n$. I used a forward approximation, *i.e.* substituting the observed mean annual values of the
reference period into Eq. (2), to estimate $\lambda_P$, $\lambda_{E_0}$, and $\lambda_n$, as did in most studies (Roderick and Farquhar,
2011; Yang and Yang, 2011; Sun *et al.*, 2014).
The decomposition method (Wang and Hejazi, 2011) calculated $\Delta R_n$ as follows:
$$\Delta R_n = R_2 - R_2^{'} = (P_2 - E_2) - (P_2 - E_2^{'}) = E_2^{'} - E_2$$
(8)

where $R_2$, $P_2$, and $E_2$ represents the mean annual runoff, precipitation and evapotranspiration of the
evaluation period; and $R_2^{'}$ and $E_2^{'}$ represents the mean annual runoff and evapotranspiration
respectively, given the climate conditions of the evaluation period and the catchment conditions of the
reference period. Both $E_2$ and $E_2^{'}$ were calculated by Eq. (1), but using $n$ values of the evaluation period
and the reference period respectively.





The complementary method (Zhou *et al.*, 2016) uses a linear combination of the complementary
relationship for runoff to determine $\Delta R_P$, $\Delta R_{E_0}$, and $\Delta R_n$ :

$$\Delta R = a\left[\left(\frac{\partial R}{\partial P}\right)_1 \Delta P + \left(\frac{\partial R}{\partial E_0}\right)_1 \Delta E_0 + P_2\Delta\left(\frac{\partial R}{\partial P}\right) + E_{0,2}\Delta\left(\frac{\partial R}{\partial E_0}\right)\right]$$
$$+(1-a)\left[\left(\frac{\partial R}{\partial P}\right)_2 \Delta P + \left(\frac{\partial R}{\partial E_0}\right)_2 \Delta E_0 + P_1\Delta\left(\frac{\partial R}{\partial P}\right) + E_{0,1}\Delta\left(\frac{\partial R}{\partial E_0}\right)\right]$$
(9)

where the subscript 1 and 2 denotes the reference and the evaluation periods, respectively. *a* is a
weighting factor and varies from 0 to 1. As suggested by Zhou *et al.* (2016), I set *a* = 0.5. Equation (9)
thus gave an estimation of $\Delta R_P$, $\Delta R_{E_0}$, and $\Delta R_n$ as follows:

$$\Delta R_P = 0.5\Delta P\left[\left(\frac{\partial R}{\partial P}\right)_1 + \left(\frac{\partial R}{\partial P}\right)_2\right]$$
(10a)

$$\Delta R_{E_0} = 0.5\Delta E_0\left[\left(\frac{\partial R}{\partial E_0}\right)_1 + \left(\frac{\partial R}{\partial E_0}\right)_2\right]$$
(10b)

$$\Delta R_n = 0.5\Delta\left(\frac{\partial R}{\partial P}\right)(P_{1+}P_2) + 0.5\Delta\left(\frac{\partial R}{\partial E_0}\right)(E_{0,1+}E_{0,2})$$
(10c)

2.5 Data
I collected data of runoff and climate of 21 selected catchments from previous studies (Table 1).
The change-point years gave in the studies was directly used to determine the reference and evaluation
periods for the LI method. As mentioned above, the LI method further divides the evaluation period into
a number of subperiods. For the sake of comparison, the last subperiod of the evaluation period was
used as the evaluation period for the decomposition, the total differential and the complementary
methods (It can be equally considered that all of the four methods used the last subperiod as the
evaluation period, but the LI method cares about the intermediate period between the reference and the
evaluation periods and the others do not). Nine of the 21 catchments had a reference period comprising
only one subperiod (Table 1), and the others had two to seven.
The 21 selected catchments were located in diverse climates and landscapes. Among them, 14
are from Australia and 7 from China (Table 1). The catchments spanned from tropical to subtropical and
temperate and from humid to semi-humid and semi-arid regions, with mean annual rainfall varying
from 506 to 1014 mm and potential evaporation from 768 to 1169 mm. The index of dryness ranges
between 0.86 and 1.91. The catchment areas vary by five orders of magnitude from 1.95 to 121,972
with a median 606 km[2]. The key data includes annual runoff, precipitation, and potential evaporation.
The record length varied between 15 and 75 with a median of 35 years. Among the 21 catchments, the
changes from the reference to the evaluation period ranged between -271 and 79 mm yr[-1] for
precipitation, and -35 and 41 mm yr[-1] for potential evaporation (Table 2). The coeval change in the
parameter *n* of the MCY equation ranged between -0.2 to 2.53. All of the catchments experienced both
climate change and land cover change over the observation period. The mean annual streamflow
reduced for all of them, by from 0.43 to 229 with a median 38 mm yr[-1]. More details of data and the
catchments can be found in Zhang *et al.* (2011), Sun *et al.* (2014), Zhang *et al.* (2010), Zheng *et al.*
(2009), Jiang *et al.* (2015), and Gao *et al.* (2016).






## 3 Results

### 3.1 Comparisons with other methods

The LI method first partitions the whole observation period into the reference and evaluation periods, then further divides the latter into a number of subperiods and evaluates the contributions to runoff from climate and catchment changes for each subperiod, and finally adds up the derived contributions. Table 3 lists all of the resultant values of $\Delta R_P$, $\Delta R_{E_0}$, and $\Delta R_n$ of the LI method, together with the three other methods.

Fig. 2(a) compares the resultant $\Delta R_n$ of the LI method and the decomposition method. Although they are quite similar, the discrepancies between them can be up to >20 mm yr$^{-1}$. The decomposition method assumes that climate change occurs first and then human interferences cause a sudden change in catchment properties. Such a fictitious path is identical to the broken line AB+BC in Fig. 1, provided that $x$ represents climate factors and $y$ catchment properties. As a result, the decomposition method can be considered as a special case of the LI method when adopting the broken line AB+BC as the integral path, as was demonstrated clearly in Fig. 2(b).

The total differentiae method is predicated on an approximate equation, *i.e.* Eq. (7). The LI method reveals that the precise form of the equation is $\Delta R = \overline{\lambda_P}\Delta P + \overline{\lambda_{E_0}}\Delta E_0 + \overline{\lambda_n}\Delta n$ (i.e. Eq. (D2) in Appendix D), where $\overline{\lambda_P}$, $\overline{\lambda_{E_0}}$ and $\overline{\lambda_n}$ (Table 4) denote the path-averaged sensitivity of $R$ to $P$, $E_0$, and $n$, respectively. All points along the path have the same weight in determining $\overline{\lambda_P}$, $\overline{\lambda_{E_0}}$ and $\overline{\lambda_n}$. To determine them, the total differential method utilizes only the initial state and the complementary method utilizes the initial and the terminal states. Neglecting the intermediate states between the initial and the terminal ones even possibly results in a reverse trend estimation (see $\Delta R_{E_0}$ for Catchment NO. 1 in Table 3). Although the elasticity method exploits information contained over the entire observation period (e.g. Zheng *et al.*, 2009; Wang *et al.*, 2013), the resultant descriptive statistics of climate elasticity may not be robust (Roderick and Farquhar, 2011; Liang *et al.*, 2015).

Superior to the total differential method, the sum of $\Delta R_P$, $\Delta R_{E_0}$, and $\Delta R_n$ always equaled to $\Delta R$ for the LI method. In addition, examination of the subperiods of the evaluation period revealed that $\partial R/\partial n$ was more temporally variable than $\partial R/\partial P$ and $\partial R/\partial E_0$ (discussed below). For this reason, $\Delta R_n$ showed considerable discrepancies between the two methods although $\Delta R_P$ as well as $\Delta R_{E_0}$ was highly correlated.

As with the LI method, the complementary method produced a $\Delta R$ on a par with the observed values. The $\Delta R_P$, $\Delta R_{E_0}$, and $\Delta R_n$ estimated by the complementary method were all in good agreement with the LI method (Fig. 4). However, the LI method often yielded values beyond the bounds given by the complementary method (Fig. 5); this is because the initial and terminal states are not equivalent to the maximum and minimum values over the integral path.





3.2 The spatio-temporal variability of the path-averaged sensitivities

$\overline{\lambda_P}$, $\overline{\lambda_{E_0}}$ and $\overline{\lambda_n}$ implies the average runoff change induced by a unit change in $P$, $E_0$ and $n$,
respectively (Appendix D). Their spatio-temporal variability is relevant to the prediction of the runoff
change. To evaluate their temporal variability, I calculated $\overline{\lambda_P}$, $\overline{\lambda_{E_0}}$ and $\overline{\lambda_n}$ for each subperiod of the
evaluation period and assessed their deviation from those for the whole evaluation period. As shown in
Fig. 6, the deviation was rather limited for $\overline{\lambda_P}$ (averaged 8.6%) and $\overline{\lambda_{E_0}}$ (averaged 13%), but was
considerable for $\overline{\lambda_n}$ (averaged 41%). Hence, it seems quite safe to predict the future climate effects on
runoff using earlier $\overline{\lambda_P}$ and $\overline{\lambda_{E_0}}$, but care must be taken when using earlier $\overline{\lambda_n}$ to predict future
catchment effect on runoff.

Different from the temporal variability, $\overline{\lambda_P}$, $\overline{\lambda_{E_0}}$ and $\overline{\lambda_n}$ all varied greatly, by up to several or
even ten folds, between the studied catchments (Table 4). It was found that there were good correlations
between $\overline{\lambda_P}$ and $P$, between $\overline{\lambda_{E_0}}$ and $P$, and between $\overline{\lambda_n}$ and $n$ (Fig. 7). Fig. 8 shows that Eq. (2)
reproduced $\overline{\lambda_P}$, $\overline{\lambda_{E_0}}$ and $\overline{\lambda_n}$ very well taking the long-term means of $P$, $E_0$, and $n$ as inputs, a fact that the
dependent variable approached its average if setting the independent variables to be their averages. The
finding would greatly facilitate the prediction of future climate effect on runoff as $\overline{\lambda_P}$ and $\overline{\lambda_{E_0}}$ was
rather stable over time as previously mentioned.

Runoff data and in turn, the parameter $n$ in the MCY equation, are often unavailable. It is thus
desirable to make predictions of $\overline{\lambda_P}$, $\overline{\lambda_{E_0}}$ and $\overline{\lambda_n}$ in the absence of the parameter $n$. I developed three
strategies as follows: 1) using Eq. (2) and assuming $n = 2$ as $n$ is typically in a small range from 1.5 to
2.6 (Roderick and Farquhar, 2011); 2) using $P$ and $E_0$ to establish regression models; 3) using the aridity
index to establish regressions as it appeared to be more correlated with both $\overline{\lambda_P}$ and $\overline{\lambda_{E_0}}$ than $P$ and $E_0$
(Fig. 7). As shown in Fig. 9, the three strategies have similar performance although the second one
seems to perform better. All of the strategies gave acceptable predictions of $\overline{\lambda_P}$ and $\overline{\lambda_{E_0}}$, but rather poor
results for $\overline{\lambda_n}$ as it was primarily controlled by $n$ ( Fig. 7). It was thus needed to seek more sophisticated
approaches to predict the future catchment effect on runoff in the absence of runoff observations.

**4 Discussion**

The LI method re-defines the widely-used concept of sensitivity at a point as the path-averaged
sensitivity. The method highlights the role of the evolutionary path in determining the resultant partition.
Yet, it seems that no studies have taken into account the path issue when evaluating the relative
influences of drivers. It has been a great concern for hydrologist, agricultural scientist, geoscientist,
catchment managers and others for more than 50 years that how much runoff change a 10% or 20%
change in precipitation would result in (Roderick and Farquhar, 2011; Yang *et al.*, 2014). The LI
method reveals that the answer to the question varies with both the timing and magnitude of the
precipitation change, not on the magnitude alone. Berghuijs and Woods (2016) claimed an asymmetry
between spatial and temporal partitioning of precipitation into streamflow and evaporation.
Unfortunately, they did not take account of the difference between the evolutionary paths over space
and time, which also play a role in determining the resultant partitioning.





Mathematically, the LI method is unrelated to a functional form and applies to communities
other than just hydrology. For example, identifying the carbon emission budgets (an allowable
amount of anthropogenic $CO_2$ emission consistent with a limiting warming target), is crucial for global
efforts to mitigate climate change. The LI method suggested that the emission budgets depends on both
the emission magnitude and pathway (timing of emissions), in line with a recent study by Gasser *et al.*
(2018), and an optimal pathway would bring about an elevated carbon budget unless the carbon-climate
system behaves in a linear fashion. The LI method applies equally to the case of spatial series of data.
Given a model that relates fluvial or aeolian sediment load to the influencing factors, for example, the
LI method can be used to separate the contributions of the factors to the sediment-load change along a
river or in the along-wind direction

**5 Conclusions**
Based on the line integral, I found a solution to partition the effects of a number of independent
variables on the change in the dependent variable. I then applied the method to partition the effects on
runoff of climatic and catchment conditions within the Budyko framework. The method reveals that in
addition to the change magnitude, the change pathways of climatic and catchment conditions also exert
control on their impacts on runoff. Instead of using the runoff sensitivity at a point, the LI method uses
the path-averaged sensitivity, thereby ensuring a mathematically precise partition. I further examined
the spatiotemporal variability of the path-averaged sensitivity. Time-wise the runoff sensitivity is stable
to climate but highly variable to catchment properties, suggesting that it is reliable to predict future
climate effects using earlier observations but care must be taken when predicting the catchment effects.
Space-wise (between catchments) the runoff sensitivity was highly variable both to climatic and
catchment conditions, but it can be well depicted by the long-term means of the climatic and catchment
conditions. As a mathematically accurate scheme, the LI method has the potential to be a generic
attribution approach in the environmental sciences.

**Data availability**
The data used in this study are freely available by contacting the authors.

**Author contribution**
MZ designed the study, analysing the data and wrote the manuscript.

**Competing interests**
The authors declare that they have no conflict of interest.





## Appendix A: Derivation of equation $\Delta z = \int_L f_x(x,\, y)dx + \int_L f_y(x,\, y)dy$


We define that the curve $L$ in Fig. 1 is given by a parametric equation: $x = x(t)$, $y = y(t)$,
$t \in [t_0, t_N]$, then $\Delta z = z_N - z_0 = f[x(t_N), y(t_N)] - f[x(t_0), y(t_0)]$. Substituting the parametric equations, we
get:
The right-hand side of the equation $= \int_{t_0}^{t_N} f_x[x(t), y(t)]dx(t) + \int_{t_0}^{t_N} f_y[x(t), y(t)]dy(t)$
$= \int_{t_0}^{t_N} \left\{ f_x[x(t), y(t)]x'(t) + f_y[x(t), y(t)]y'(t) \right\} dt$           (A1)

Let $g(t) = f[x(t), y(t)]$, and after using the chain rule to differentiate $g$ with respect to $t$, we obtain:

$g'(t) = \dfrac{\partial g}{\partial x}\dfrac{dx}{dt} + \dfrac{\partial g}{\partial y}\dfrac{dy}{dt} = f_x[x(t), y(t)]x'(t) + f_y[x(t), y(t)]y'(t)$      (A2)

It shows that $g'(t)$ is just the integrand in Eq. (A1), Eq. (A1) can then be rewritten as:

The     right-hand     side     of     the     equation     $= \int_{t_0}^{t_N} g'(t)\, dt = \left[ g(t) \right]_{t_0}^{t_N} = g(t_N) - g(t_0)$
$= f[x(t_N), y(t_N)] - f[x(t_0), y(t_0)] =$ The left-hand side of the equation

## Appendix B: The sum of $\int_L f_x(x,\, y)dx$ and $\int_L f_y(x,\, y)dy$ is path independent


**Theorem**: Given an open simply-connected region $G$ (i.e., no holes in $G$) and two functions $P(x, y)$
and $Q(x, y)$ that have continuous first-order derivatives, if and only if $\partial P / \partial y = \partial Q / \partial x$ throughout $G$,
then $\int_L P(x, y)dx + \int_L Q(x, y)dy$ is path independent, i.e., it depends solely on the starting and ending
point of $L$.

We have $\partial f_x / \partial y = \partial^2 z / \partial x \partial y$ and $\partial f_y / \partial x = \partial^2 z / \partial y \partial x$. As $\partial^2 z / \partial x \partial y = \partial^2 z / \partial y \partial x$, we can state that
$\partial f_x / \partial y = \partial f_y / \partial x$, meeting the above condition and proving that $\int_L f_x(x, y)dx + \int_L f_y(x, y)dy$ is path
independent. The statement was further exemplified using a fictitious example in Appendix C.

## Appendix C. A fictitious example to show how the LI method works


It is assumed that runoff ($R$, mm yr$^{-1}$) at a site increases from 120 to 195 mm yr$^{-1}$ with $\Delta R = 75$ mm
yr$^{-1}$; meanwhile, precipitation ($P$, mm yr$^{-1}$) varies from 600 to 650 mm yr$^{-1}$ ($\Delta P = 75$ mm yr$^{-1}$) and
runoff coefficient ($C_R$, dimensionless) from 0.2 to 0.3 ($\Delta C_R = 0.1$). The goal is to partition $\Delta R$ into the
effects of precipitation ($\Delta R_P$) and runoff coefficient ($\Delta R_{C_R}$) provided that $P$ and $C_R$ are independent.
We have a function $R = PC_R$ and its partial derivatives $\partial R / \partial P = C_R$ and $\partial R / \partial C_R = P$. Given a path $L$
along which $P$ and $C_R$ change and using Eq. (3), the LI method evaluates $\Delta R_P$ and $\Delta R_{C_R}$ as:

$\Delta R_{C_R} = \int_L \partial R / \partial C_R dC_R = \int_L P dC_R$ and $\Delta R_P = \int_L \partial R / \partial P dP = \int_L C_R dP$     (C1)





The result differs depending on $L$ but the sum of $\Delta R_P$ and $\Delta R_{C_R}$ uniformly equals $\Delta R$. It will be
demonstrated using Fig. 1, in which the $x$-axis represents $C_R$ and the $y$-axis $P$. Point A denotes the initial
state ($C_R = 0.2$, $P = 600$) and point C the terminal state ($C_R = 0.3$, $P = 650$). I calculated $\Delta R_P$ and $\Delta R_{C_R}$
along three fictitious paths as follows:
1)    $L$=AC. Line segment AC has equation $P = 500C_R + 500, 0.2 \leq C_R \leq 0.3$. Let's take $C_R$ as the
parameter and write the equation in the parametric form as $P = 500C_R + 500, C_R = C_R, 0.2 \leq C_R \leq 0.3$.
By substituting the equation into Eq. (C1), we have:
$$\Delta R_{C_R} = \int_{AC} P dC_R = \int_{0.2}^{0.3} (500C_R + 500) dC_R = 62.5$$
$$\Delta R_P = \int_{AC} C_R dP = \int_{AC} C_R d(500C_R + 500) = 500 \int_{0.2}^{0.3} C_R dC_R = 12.5$$
2) $L$=AB+BC. To evaluate on the broken line, we can evaluate separately on AB and BC and then sum
them up. The equation for AB is $P = 600, 0.2 \leq C_R \leq 0.3$, and is $C_R = 0.3, 600 \leq P \leq 650$ for BC.
Notes that a constant $C_R$ or $P$ implies that $dC_R = 0$ or $dP = 0$. Eq. (C1) then becomes:
$$\Delta R_{C_R} = \int_{AB+BC} P dC_R = \int_{AB} P dC_R + \int_{BC} P dC_R = \int_{0.2}^{0.3} 600 dC_R + 0 = 60$$
$$\Delta R_P = \int_{AB+BC} C_R dP = \int_{AB} C_R dP + \int_{BC} C_R dP = 0 + \int_{600}^{650} 0.3 dP = 15$$
3)    $L$=AD+DC. The equation for AD is $C_R = 0.2, 600 \leq P \leq 650$ and is $P = 650, 0.2 \leq C_R \leq 0.3$ for
DC. $\Delta R_P$ and $\Delta R_{C_R}$ are evaluated as:
$$\Delta R_{C_R} = \int_{AD+DC} P dC_R = \int_{AD} P dC_R + \int_{DC} P dC_R = 0 + \int_{0.2}^{0.3} 650 dC_R = 65$$
$$\Delta R_P = \int_{AD+DC} C_R dP = \int_{AD} C_R dP + \int_{DC} C_R dP = \int_{600}^{650} 0.2 dP + 0 = 10$$
As we expected, the sum of $\Delta R_P$ and $\Delta R_{C_R}$ persistently equals $\Delta R$ although $\Delta R_P$ and $\Delta R_{C_R}$ varies with
$L$.

**Appendix D: Derivation of** $\Delta R = \overline{\lambda_P} \Delta P + \overline{\lambda_{E_0}} \Delta E_0 + \overline{\lambda_n} \Delta n$
If we partition the interval $[x_0, x_N]$ in Fig. 1 into $N$ distinct bins of the same width $\Delta x_i = \Delta x / N$. Eq.
(3a) can then be rewritten as:
$$\Delta Z_x = \int_L f_x(x, y) dx = \lim_{\tau \to 0} \sum_{i=0}^{N-1} f_x(x_i, y_i) \Delta x_i = \lim_{N \to \infty} N \Delta x_i \frac{\sum_{i=0}^{N-1} f_x(x_i, y_i)}{N} = \Delta x \lim_{N \to \infty} \frac{\sum_{i=0}^{N-1} f_x(x_i, y_i)}{N} = \overline{\lambda_x} \Delta x$$
where $\overline{\lambda_x} = \lim_{N \to \infty} \dfrac{\sum_{i=1}^{N} f_x(x_i, y_i)}{N}$, denoting the average of $f_x(x, y)$ along the curve $L$. Likewise, we have
$\Delta Z_y = \overline{\lambda_y} \Delta y$, where $\overline{\lambda_y}$ denotes the average of $f_y(x, y)$ along the curve $L$. As a result, we have:
$\Delta Z = \overline{\lambda_x} \Delta x + \overline{\lambda_y} \Delta y$                    (D1)





The result can readily be extended to a function of three variables. Applying the mathematic
derivation above to the MCY Equation results in a precise form of Eq. (7):
$\Delta R = \Delta R_P + \Delta R_{E0} + \Delta R_n = \overline{\lambda_P}\Delta P + \overline{\lambda_{E0}}\Delta E_0 + \overline{\lambda_n}\Delta n$,         (D2)
where $\Delta R_P = \overline{\lambda_P}\Delta P$, $\Delta R_{E0} = \overline{\lambda_{E0}}\Delta E_0$, $\Delta R_n = \overline{\lambda_n}\Delta n$, and $\overline{\lambda_P}$, $\overline{\lambda_{E0}}$ and $\overline{\lambda_n}$ denote the arithmetic mean of $\partial R/\partial P$,
$\partial R/\partial E_0$, and $\partial R/\partial n$ along a path of climate and catchment changes, respectively. Because $\overline{\lambda_P} = \Delta R_P / \Delta P$,
$\overline{\lambda_{E0}} = \Delta R_{E0} / \Delta E_0$, and $\overline{\lambda_n} = \Delta R_n / \Delta n$, $\overline{\lambda_P}$, $\overline{\lambda_{E0}}$ and $\overline{\lambda_n}$ also implies the runoff change due to a unit change in
$P$, $E_0$ and $n$, respectively.

## Acknowledgments

This work was funded by the National Natural Science Foundation of China (41671278), the GDAS'
Project of Science and Technology Development (2019GDASYL-0103043) and (2019GDASYL-
0502004). I thank Mr. Y.Q. Zheng for his assistance with the mathematic derivations.

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



**Table 1.** Summary of the long-term hydrometeorological characteristics of the selected catchments[a]

| Catchment No.[b] | Area (km$^2$) | $R$ | $P$ | $E_0$ | $n$ | $AI$ | Reference Period | Evaluation Period | The Last Subperiod |
|---|---|---|---|---|---|---|---|---|---|
| 1 | 391 | 218 | 1014 | 935 | 3.5 | 0.92 | 1933-1955 | 1956-2008 | 1998-2008 |
| 2 | 16.64 | 32.9 | 634 | 1087 | 3.16 | 1.71 | 1979-1984 | 1985-2008 | 1999-2008 |
| 3 | 559 | 183 | 787 | 780 | 2.68 | 0.99 | 1960-1978 | 1979-2000 | 1993-2000 |
| 4 | 606 | 73 | 729 | 998 | 3.07 | 1.37 | 1971-1995 | 1996-2009 | 2003-2009 |
| 5 | 760 | 77.9 | 689 | 997 | 2.66 | 1.45 | 1970-1995 | 1996-2009 | 2003-2009 |
| *6* | 502 | 57.2 | 730 | 988 | 3.59 | 1.35 | 1974-1995 | *1996-2008* | *1996-2008* |
| 7 | 673 | 431 | 1013 | 953 | 1.34 | 0.94 | 1947-1955 | 1956-2008 | 1998-2008 |
| 8 | 390 | 139 | 840 | 1021 | 2.61 | 1.22 | 1966-1980 | 1981-2005 | 1995-2005 |
| 9 | 1130 | 20.7 | 633 | 1077 | 3.79 | 1.7 | 1972-1982 | 1983-2007 | 1997-2007 |
| 10 | 3.2 | 37.5 | 631 | 954 | 3.49 | 1.51 | 1989-1991 | 1992-2009 | 1999-2009 |
| *11* | 1.95 | 111 | 767 | 901 | 3.06 | 1.18 | 1990-1992 | *1993-2005* | *1993-2005* |
| 12 | 89 | 272 | 963 | 826 | 2.82 | 0.86 | 1958-1965 | 1966-1999 | 1987-1999 |
| *13* | 243 | 38.5 | 735 | 1010 | 4.27 | 1.37 | 1989-1995 | *1996-2007* | *1996-2007* |
| *14* | 56.35 | 65.8 | 744 | 1007 | 3.35 | 1.35 | 1989-1995 | *1996-2008* | *1996-2008* |
| *15* | 14484 | 385 | 893 | 1022 | 1.11 | 1.14 | 1970-1989 | *1990-2000* | *1990-2000* |
| *16* | 38625 | 461 | 985 | 1087 | 1.03 | 1.1 | 1970-1989 | *1990-2000* | *1990-2000* |
| *17* | 59115 | 388 | 897 | 1161 | 1.02 | 1.29 | 1970-1989 | *1990-2000* | *1990-2000* |
| *18* | 95217 | 371 | 881 | 1169 | 1.03 | 1.33 | 1970-1989 | *1990-2000* | *1990-2000* |
| *19* | 121,972 | 171 | 507 | 768 | 1.17 | 1.52 | 1960-1990 | *1991-2000* | *1991-2000* |
| 20 | 106,500 | 60.5 | 535 | 905 | 2.25 | 1.69 | 1960-1970 | 1971-2009 | 1999-2009 |
| 21 | 5891 | 34.4 | 506 | 964 | 2.54 | 1.91 | 1952-1996 | 1997-2011 | 2004-2011 |

[a]$R$, $P$, and $E_0$ represents mean annual runoff, precipitation and potential evaporation, all in mm yr$^{-1}$. $n$ (dimensionless) is the parameter representing catchment properties in the MCY equation. $AI$ is dimensionless aridity index ($AI = E_0/P$). Data of Catchments 1-14 were derived from Zhang *et al.* (2010). Data of Catchments 15-18 were from Sun *et al.* (2014). Data of Catchments 19-21 were from Zheng *et al.* (2009), Jiang *et al.* (2015), and Gao *et al.* (2016), respectively. I used the change points given in the literatures to divide the observation period into the reference and elevation periods. The LI method further divides the evaluation period into a number of subperiods. The column "The Last Subperiod" denotes the last one, which was used as the evaluation period for the total differential method, the decomposition method and the complementary method. The bold and italic rows denote that the column "Evaluation Period" is the same as the column "The Last Subperiod".

[b]Catchments 1-14 are in Ausralia and the others in China. 1: Adjungbilly CK; 2: Batalling Ck; 3: Bombala River; 4: Crawford River; 5: Darlot Ck; 6: Eumeralla River; 7: Goobarragandra CK; 8: Jingellic CK; 9: Mosquito CK; 10: Pine Ck; 11: Red Hill; 12: Traralgon Ck; 13: Upper Denmark River; 14: Yate Flat Ck; 15: Yangxian station, Hang River; 16: Ankang station, Hang River; 17: Baihe station, Hang River; 18: Danjiangkou station, Hang River; 19: Headwaters of the Yellow River Basin; 20: Wei River; 21: Yan River.





552

553

**Table 2.** Comparisons of $R$ (mm yr$^{-1}$), $P$ (mm yr$^{-1}$), $E_0$ (mm yr$^{-1}$), and $n$ (dimensionless) between the reference and the evaluation periods[a]

| Catchment No. | $R_1$ | $R_2$ | $P_1$ | $P_2$ | $E_{01}$ | $E_{02}$ | $n_1$ | $n_2$ | $\Delta R$ | $\Delta P$ | $\Delta E_0$ | $\Delta n$ |
|---|---|---|---|---|---|---|---|---|---|---|---|---|
| 1 | 223 | 216 | 959 | 1038 | 950 | 928 | 2.7 | 4.1 | -7.2 | 79.2 | -21 | 1.4 |
| 2 | 40.6 | 31 | 655 | 629 | 1087 | 1087 | 3 | 3.2 | -9.7 | -27 | 0 | 0.2 |
| 3 | 249 | 127 | 847 | 736 | 780 | 780 | 2.3 | 3.2 | -122 | -112 | 0.4 | 0.9 |
| 4 | 90.6 | 41.5 | 753 | 685 | 1002 | 989 | 2.9 | 3.7 | -49 | -67 | -13 | 0.8 |
| 5 | 94.9 | 46.3 | 718 | 633 | 1000 | 992 | 2.5 | 3 | -49 | -85 | -9 | 0.5 |
| 6 | 70.8 | 34.3 | 756 | 687 | 989 | 987 | 3.4 | 4.1 | -36 | -69 | -2 | 0.6 |
| 7 | 575 | 406 | 1123 | 995 | 931 | 957 | 1.1 | 1.4 | -169 | -128 | 25 | 0.3 |
| 8 | 139 | 139 | 871 | 821 | 1043 | 1008 | 2.7 | 2.5 | -0.4 | -50 | -35 | 0 |
| 9 | 24.1 | 19.2 | 659 | 621 | 1100 | 1067 | 3.7 | 3.8 | -4.9 | -37 | -33 | 0.1 |
| 10 | 116 | 24.3 | 588 | 638 | 927 | 958 | 1.7 | 4.2 | -92 | 50.4 | 31 | 2.5 |
| 11 | 297 | 68 | 986 | 716 | 884 | 905 | 2.3 | 3.6 | -229 | -271 | 22 | 1.3 |
| 12 | 301 | 265 | 992 | 956 | 820 | 828 | 2.7 | 2.8 | -36 | -36 | 7.4 | 0.1 |
| 13 | 48.5 | 32.6 | 752 | 725 | 991 | 1021 | 4.2 | 4.4 | -16 | -28 | 30 | 0.2 |
| 14 | 90.4 | 52.6 | 753 | 739 | 991 | 1015 | 2.9 | 3.7 | -38 | -14 | 24 | 0.8 |
| 15 | 435 | 295 | 948 | 795 | 1008 | 1047 | 1.1 | 1.2 | -139 | -153 | 38 | 0.1 |
| 16 | 520 | 353 | 1035 | 894 | 1074 | 1109 | 1 | 1.2 | -167 | -141 | 35 | 0.2 |
| 17 | 441 | 291 | 939 | 820 | 1149 | 1182 | 1 | 1.2 | -151 | -119 | 33 | 0.2 |
| 18 | 412 | 296 | 913 | 821 | 1163 | 1179 | 1 | 1.1 | -116 | -92 | 15 | 0.2 |
| 19 | 180 | 144 | 512 | 491 | 774 | 751 | 1.1 | 1.3 | -36 | -21 | -23 | 0.2 |
| 20 | 90.2 | 52.1 | 585 | 520 | 895 | 908 | 2.1 | 2.3 | -38 | -65 | 13 | 0.2 |
| 21 | 37.7 | 24.6 | 521 | 462 | 954 | 995 | 2.6 | 2.5 | -13 | -59 | 41 | 0 |

[a]The subscript "1" denotes the reference period and "2" denotes the evaluation period. $\Delta X = X_2 - X_1$ ($X$ as a substitute for $R$, $P$, $E_0$, and $n$).





**Table 3.** Effects of precipitation ($\Delta R_P$, mm yr$^{-1}$), potential evapotranspiration ($\Delta R_{E_0}$, mm yr$^{-1}$), and catchment ($\Delta R_n$, mm yr$^{-1}$) changes on the mean annual runoff resulting from the four methods

| Catchment NO.[a] | LI Method | | | Decomposition Method | Total Differential Method | | | Complementary Method | | |
|---|---|---|---|---|---|---|---|---|---|---|
| | $\Delta R_P$ | $\Delta R_{E_0}$ | $\Delta R_n$ | $\Delta R_n$ | $\Delta R_P$ | $\Delta R_{E_0}$ | $\Delta R_n$ | $\Delta R_P$ | $\Delta R_{E_0}$ | $\Delta R_n$ |
| 1 | -70.9 | -8.99 | -24.3 | -44.6 | -67 | 4.82 | -62 | -60.7 | 4.34 | -47.3 |
| 2 | -6.49 | 0.95 | -9.74 | -9.65 | -7.2 | 1.3 | -13 | -6.23 | 1.13 | -10.2 |
| 3 | -89 | 25.9 | -140 | -128 | -104 | 26.6 | -483 | -88 | 25.7 | -140 |
| 4 | -18.1 | 2.09 | -35.4 | -36.3 | -18 | 2.37 | -58 | -14.8 | 1.99 | -38.5 |
| 5 | -27.9 | 1.14 | -21.3 | -18.6 | -34 | 1.18 | -27 | -28.1 | 0.97 | -20.9 |
| *6* | -19.9 | 0.29 | -16.7 | -14.9 | -24 | 0.36 | -22 | -19.9 | 0.29 | -16.7 |
| 7 | -211 | -7.19 | -101 | -90.9 | -236 | -6.9 | -134 | -211 | -6.21 | -102 |
| 8 | -32.2 | 12.3 | -14.4 | -12.6 | -35 | 12.6 | -15 | -32.9 | 11.9 | -13.3 |
| 9 | -11.8 | 3.02 | -9.96 | -8.45 | -13 | 0.85 | -20 | -8.76 | 0.56 | -10.5 |
| 10 | 19.47 | -5.61 | -119 | -96.5 | 0.91 | -10 | -291 | 0.56 | -6.53 | -99.1 |
| *11* | -150 | -7.46 | -71.8 | -60.7 | -188 | -9.4 | -113 | -144 | -7.04 | -78.3 |
| 12 | -9.88 | -3.99 | -79.2 | -82 | -11 | -0.5 | -154 | -10.8 | -0.57 | -81.6 |
| *13* | -6.98 | -4.36 | -4.54 | -4.21 | -8 | -5.1 | -5.2 | -7 | -4.38 | -4.51 |
| *14* | -4.84 | -4.42 | -28.7 | -27.9 | -5.6 | -5 | -37 | -4.85 | -4.4 | -28.6 |
| *15* | -104 | -8.56 | -24.8 | -23 | -110 | -9.4 | -27 | -103 | -8.52 | -25.1 |
| *16* | -99.3 | -7.99 | -58.8 | -56 | -105 | -8.3 | -68 | -99 | -7.92 | -59.1 |
| *17* | -78.8 | -6.26 | -63.9 | -61 | -84 | -6.5 | -76 | -78.6 | -6.2 | -64.2 |
| *18* | -60.1 | -2.79 | -53.5 | -52 | -64 | -2.9 | -62 | -60 | -2.77 | -53.6 |
| *19* | -11.9 | 3.89 | -27.6 | -27 | -12 | 3.81 | -31 | -11.9 | 3.85 | -27.5 |
| 20 | -27.5 | -2.46 | -18.5 | -17 | -31 | -4.4 | -26 | -25.5 | -3.47 | -19.5 |
| 21 | -10.4 | -3.47 | -2.11 | -3.4 | -9.9 | -4.8 | -4.8 | -8.27 | -3.86 | -3.82 |

[a]The bold and italic numbers denote that the evaluation period of the catchment comprised a single subperiod.





592

**Table 4.** Comparisons of the path-averaged with the point sensitivities of runoff [a, b]

| Catchment NO. | $\overline{\lambda_P}$ | $\overline{\lambda_{E0}}$ | $\overline{\lambda_n}$ | $\lambda_{Pf}$ | $\lambda_{E0f}$ | $\lambda_{nf}$ | $\lambda_{Pb}$ | $\lambda_{E0b}$ | $\lambda_{nb}$ |
|---|---|---|---|---|---|---|---|---|---|
| 1 | 0.68 | -0.55 | -17 | 0.621 | -0.39 | -71.8 | 0.497 | -0.32 | -39.7 |
| 2 | 0.2 | -0.08 | -27.3 | 0.227 | -0.1 | -30.9 | 0.168 | -0.07 | -19.6 |
| 3 | 0.58 | -0.36 | -26.7 | 0.68 | -0.42 | -79 | 0.473 | -0.39 | -6.29 |
| 4 | 0.3 | -0.16 | -30.5 | 0.39 | -0.2 | -50.1 | 0.248 | -0.14 | -21 |
| 5 | 0.33 | -0.14 | -43.1 | 0.394 | -0.19 | -59.4 | 0.264 | -0.12 | -33.2 |
| 6 | 0.29 | -0.16 | -26.5 | 0.352 | -0.2 | -34.9 | 0.228 | -0.12 | -19.1 |
| 7 | 0.71 | -0.32 | -223 | 0.781 | -0.33 | -299 | 0.615 | -0.26 | -157 |
| 8 | 0.49 | -0.26 | -77.9 | 0.478 | -0.27 | -64.9 | 0.429 | -0.24 | -50.7 |
| 9 | 0.16 | -0.07 | -11.8 | 0.161 | -0.07 | -17.6 | 0.052 | -0.02 | -4.31 |
| 10 | 0.27 | -0.12 | -40.9 | 0.45 | -0.16 | -99.9 | 0.101 | -0.05 | -7.8 |
| 11 | 0.55 | -0.35 | -56.1 | 0.695 | -0.44 | -88.2 | 0.367 | -0.22 | -30.7 |
| 12 | 0.72 | -0.45 | -57.3 | 0.74 | -0.53 | -61.1 | 0.775 | -0.67 | -16.7 |
| 13 | 0.25 | -0.15 | -19.8 | 0.29 | -0.17 | -22.5 | 0.219 | -0.12 | -17.1 |
| 14 | 0.34 | -0.18 | -37.2 | 0.393 | -0.21 | -48.6 | 0.291 | -0.16 | -27.8 |
| 15 | 0.68 | -0.22 | -275 | 0.719 | -0.25 | -303 | 0.635 | -0.2 | -246 |
| 16 | 0.7 | -0.23 | -326 | 0.745 | -0.24 | -378 | 0.659 | -0.21 | -279 |
| 17 | 0.66 | -0.19 | -320 | 0.708 | -0.2 | -378 | 0.609 | -0.18 | -267 |
| 18 | 0.65 | -0.19 | -315 | 0.692 | -0.19 | -363 | 0.614 | -0.18 | -270 |
| 19 | 0.58 | -0.17 | -153 | 0.602 | -0.17 | -175 | 0.552 | -0.17 | -134 |
| 20 | 0.32 | -0.12 | -50.1 | 0.402 | -0.16 | -69.6 | 0.255 | -0.1 | -37.7 |
| 21 | 0.2 | -0.06 | -29.2 | 0.234 | -0.09 | -34 | 0.157 | -0.05 | -22.6 |

[a] $\overline{\lambda_P}$ (mm mm$^{-1}$), $\overline{\lambda_{E0}}$ ( mm mm$^{-1}$), and $\overline{\lambda_n}$ (dimensionless) represent the path-averaged sensitivities of runoff to precipitation, potential evaporation, and catchment properties. If the evaluation period comprises only one subperiod, $\overline{\lambda_P}$ , $\overline{\lambda_{E0}}$ and $\overline{\lambda_n}$ was calculated as: $\overline{\lambda_P} = \Delta R_P / \Delta P$ , $\overline{\lambda_{E0}} = \Delta R_{E0} / \Delta E_0$ ,and $\overline{\lambda_n} = \Delta R_n / \Delta n$ . If the evaluation period comprises $N>1$ subperiods, the equations become: $\overline{\lambda_P} = \sum_{i=1}^{N} |\Delta R_{Pi}| / \sum_{i=1}^{N} |\Delta P_i|$ , $\overline{\lambda_{E0}} = -\sum_{i=1}^{N} |\Delta R_{E0i}| / \sum_{i=1}^{N} |\Delta E_{0i}|$ , and $\overline{\lambda_n} = -\sum_{i=1}^{N} |\Delta R_{ni}| / \sum_{i=1}^{N} |\Delta n_i|$ , where the subscript $i$ denotes the $i$th subperiod.

[b] $\lambda_P$ , $\lambda_{E0}$ , and $\lambda_n$ represent the point sensitivities of runoff . The subscript "$f$" represents a forward approximation, i.e. substituting the observed mean annual values of the reference period into Eq. (2) to calculate the sensitivities, while the subscript "$b$" represents a backward approximation (Zhou *et al.*, 2016), *i.e.* substituting the observed mean annual values of the evaluation period into Eq. (2).

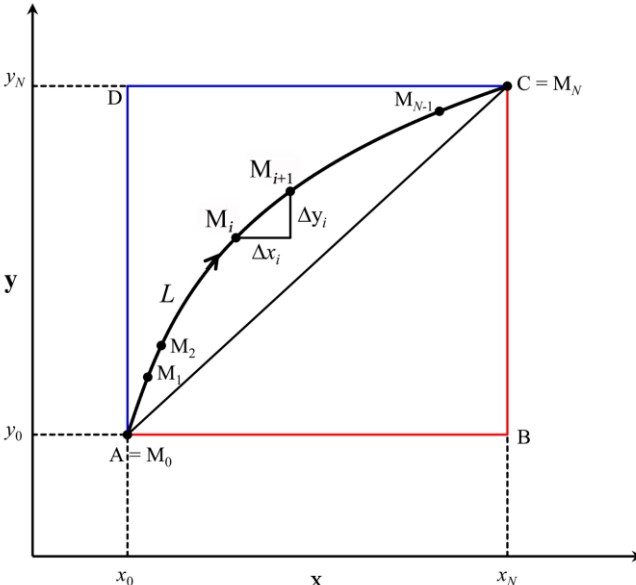

**Fig. 1**. A schematic plot to illustrate the LI method.

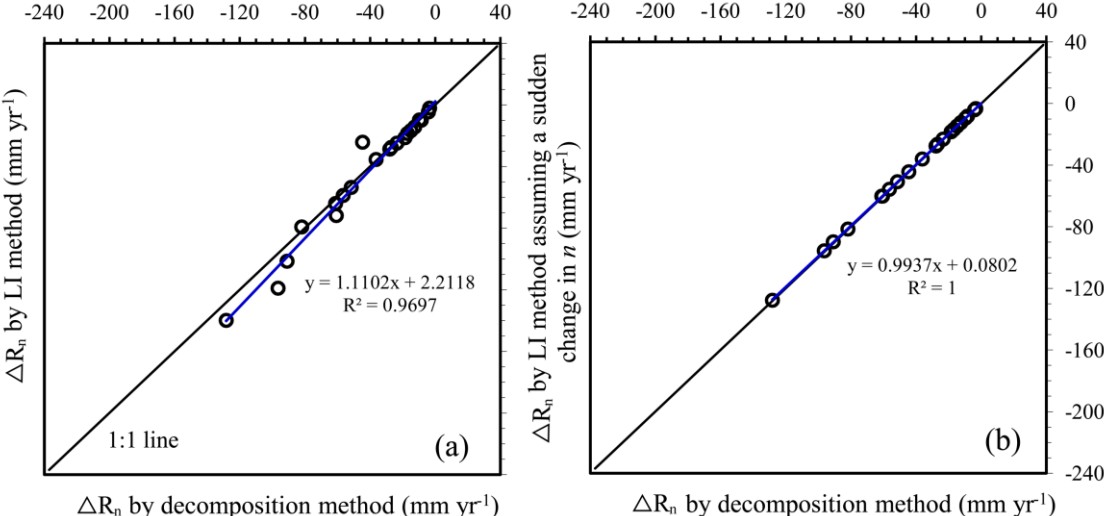

**Fig. 2**. Comparison between the LI method and the decomposition method. (a) Comparison of the
estimated contribution to the runoff change from catchment change ($\Delta R_n$); (b) the decomposition
method is equivalent to the LI method that assumes a sudden change in catchment properties following
climate change. In this case, the integral path of the LI method is the broken line AB+BC in Fig. 1 ($x$
represents climate factors and $y$ catchment properties, *i.e.* $n$) and
$$\Delta R_n = \int_{AB+BC} \frac{\partial R}{\partial n} dn = \int_{AB} \frac{\partial R}{\partial n} dn + \int_{BC} \frac{\partial R}{\partial n} dn = 0 + \int_{BC} \frac{\partial R}{\partial n} dn = \int_{n_1}^{n_2} f_n(P_2, E_{02}, n) dn \quad , \quad \text{where the subscript "1"}$$
denotes the reference period and "2" denotes the last subperiod of the evaluation period.


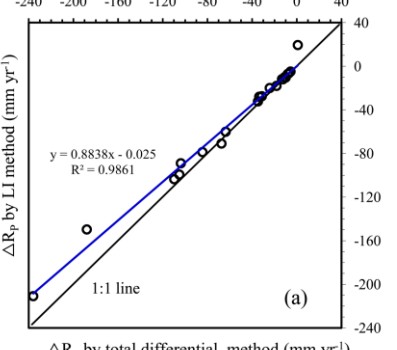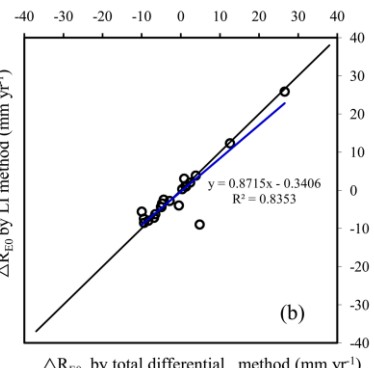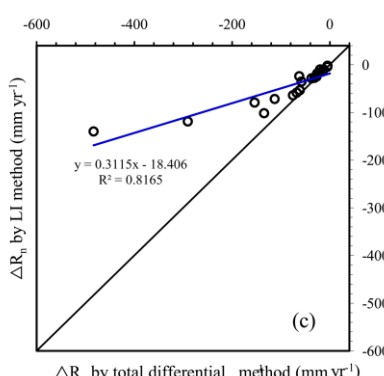

**Fig. 3**. Comparison of the estimated contribution to runoff from the changes in (a) precipitation ($\Delta R_P$), (b) potential evapotranspiration ($\Delta R_{E0}$), and (c) catchment properties ($\Delta R_n$) between the LI method and the total differential method.

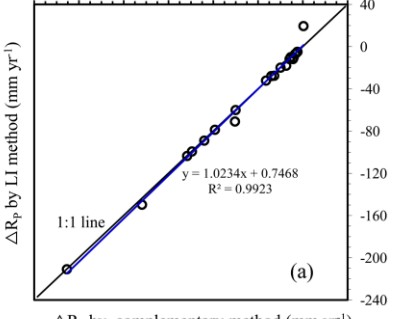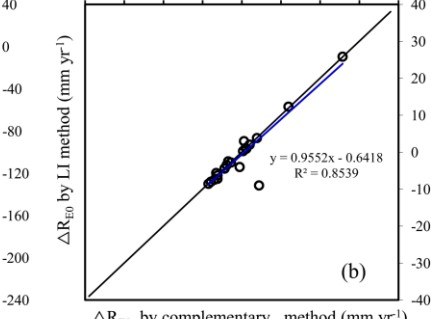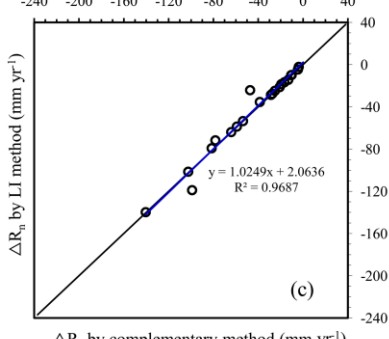

**Fig. 4**. Comparison of (a) $\Delta R_P$, (b) $\Delta R_{E0}$, and (c) $\Delta R_n$ between the LI method and the complementary method ($a = 0.5$).

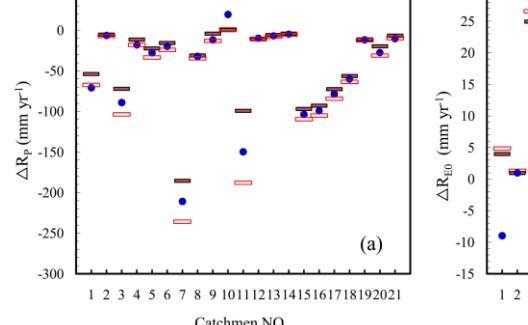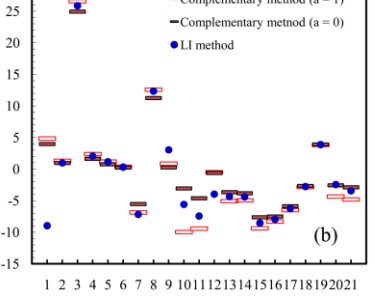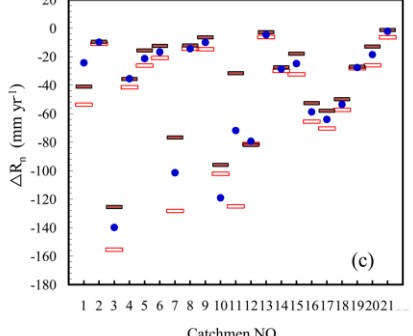

**Fig. 5**. Comparison of (a) $\Delta R_P$, (b) $\Delta R_{E0}$, and (c) $\Delta R_n$ by the LI method with the upper ($a=1$) and lower ($a=0$) bounds given by the complementary method.

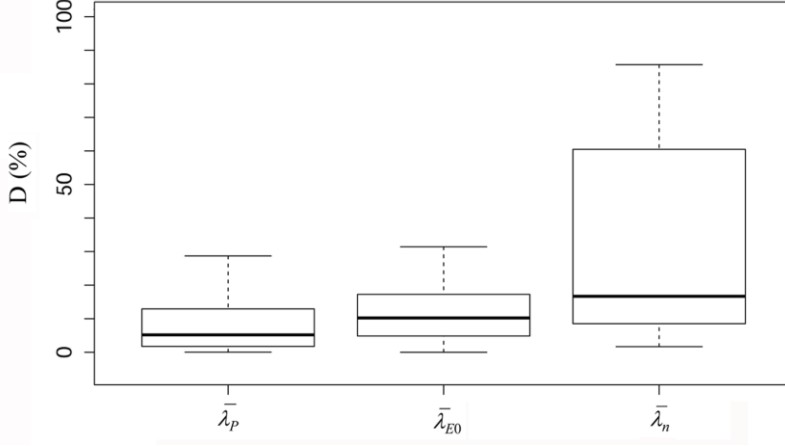

**Fig. 6**. Boxplots showing the temporal variability of the path-averaged sensitivities of water yield to precipitation ($\overline{\lambda_P}$), potential evapotranspiration ($\overline{\lambda_{E0}}$), and catchment properties ($\overline{\lambda_n}$). $D$ (%) was calculated as the relative difference between the sensitivity of the whole evaluation period and that of a subperiod. In the calculations, I excluded the catchments whose evaluation periods were not long enough to comprise two or more subperiods. Box spans the inter-quartile range (IQR) and solid lines are medians. Whiskers represent data range, excluding statistical outliers, which extend more than 1.5IQR from the box ends.


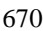

670

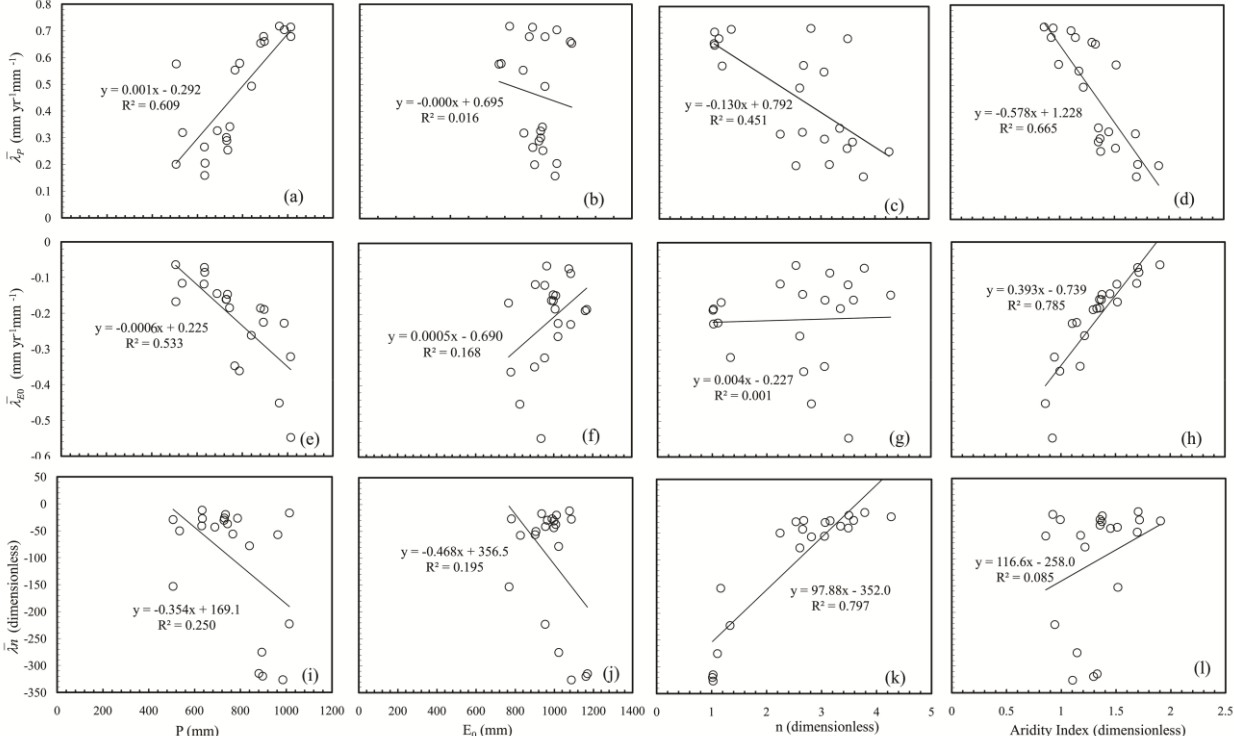

671

**Fig. 7**. $\overline{\lambda_P}$, $\overline{\lambda_{E_0}}$ and $\overline{\lambda_n}$ in correlation with $P$, $E_0$, $n$, and aridity index.

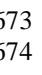
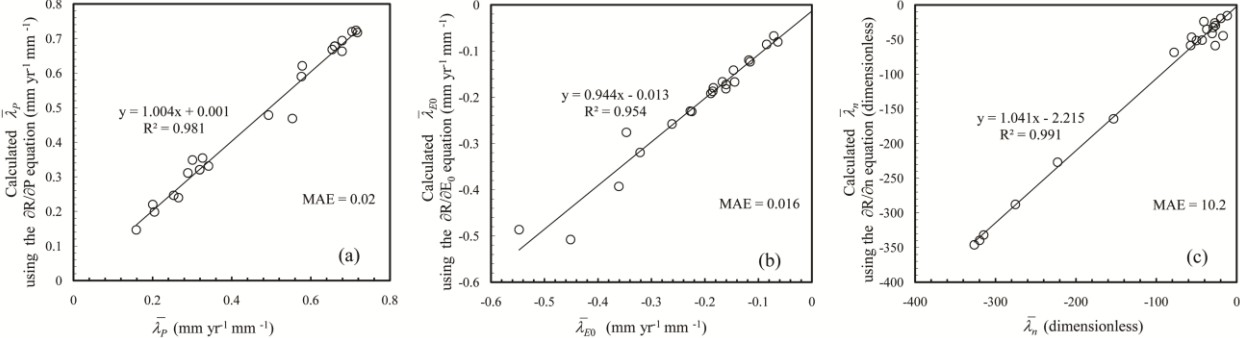

675

**Fig. 8**. Comparisons of $\overline{\lambda_P}$, $\overline{\lambda_{E_0}}$ and $\overline{\lambda_n}$ (given in Table 4) with those predicted using Eq. (2) with the long-term mean values of $P$, $E_0$, and $n$ as inputs. $MAE = N^{-1}\sum_{i=1}^{N}\left|O_i - P_i\right|$, is the mean absolute error, where $O$ and $P$ are values that actually encountered (given in Table 4) and predicted using Eq. (2) respectively, and $N$ is the number of selected catchments.

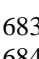

683
684



685

**Fig. 9.** Comparisons of $\overline{\lambda_P}$, $\overline{\lambda_{E_0}}$ and $\overline{\lambda_n}$ with those predicted by the three strategies. (a)-(c) by Eq. (2) with a constant $n$ ($n = 2$), (d)-(f) by the regression equations established using $P$ and $E_0$: $\overline{\lambda_P} = 0.0011P - 0.0006E_0 + 0.21$ ($R^2 = 0.7$), $\overline{\lambda_{E_0}} = 0.0007P - 0.0007E_0 - 0.38$ ($R^2 = 0.87$), and $\overline{\lambda_n} = -0.302P - 0.372E_0 + 493$ ($R^2$ 0.37), and (g)-(i) by the regression equations established using only the aridity index, as shown in Fig. 7 (d), (h) and (l). *MAE* was calculated as for Fig. 8.

691

692

693
694