# Peer review of "A line integral-based method to partition climate and catchment effects on runoff"

_Hydrology and Earth System Sciences, 2019_

## Referee Comment (RC1) · Anonymous Referee #1 · 24 Oct 2019

I have selected "Good" for scientific significance and quality.

If you see my report you realise that I am not saying that. The system would not let me submit a NULL answer.

For this manuscript I have a NULL answer for all of the above - see the report.

Please also note the supplement to this comment:
https://www.hydrol-earth-syst-sci-discuss.net/hess-2019-452/hess-2019-452-RC1-supplement.pdf

———————————————

[Figure]

**Supplement:**

**Review of HESS Manuscript #2019-452**

Journal: HESS
Title: A line integral-based method to partition climate and catchment effects on runoff
Author(s): Mingguo Zheng
MS No.: hess-2019-452
MS Type: Research article

**Review**

This manuscript describes mathematical research. The application is to a classical hydrological problem but the results are about comparing (theoretical) calculated quantities.

In more detail, the formulation of the problem addressed in described in detail on pages 2-3 (lines 77-89). The basic idea is that the standard first order expansion for a total differential does not adequately consider the order of the differentiation.

A new proposal is made that enables the first order expansion to be used. In short I did not understand the proposed formulation of the problem.

To my mind this is classical calculus and it may be better to get a professional mathematician to evaluate the work. My own evaluation is that I could not see the underlying point of the formulation. On my understanding (and remembering that I am not a professional mathematician) we use a first order expansion to get the total differential, and each of the individual differentials are considered to be infinitesimal in which it does not matter about the order. If we want more detail then we make a second order expansion, e.g., using the example from the text, i.e., R=f(x, y), we have for the relevant second order term a differential like;
$\partial^2 R/(\partial x \, \partial y)$
to more fully account for the missing part. Such rigour is rarely used in Hydrological (or science) practice since we usually have finite differences (rather than differentials) and the necessary accuracy is usually only 10% or so.

Have I missed something important?

**Recommend: Not much I can say here.**

---

## Referee Comment (RC2) · Anonymous Referee #2 · 18 Nov 2019

The paper describes a mathematical method to attribute a discrete change in runoff to changes in climate and catchment characteristics. The method is directly applicable to common data and yields quite similar results when compared to existing methods. However, it remains open which of these methods is more accurate because there is no data to verify.

Still, there are two interesting and valuable aspects of the manuscript: a) The role of the evolution over time b) Reconciling the existing methods and their assumptions on this evolution

To consider the path of changes is an important aspect and, as the author illustrates, may thus alter the resultant sensitivity to a change. This is important, since this may allow to better assess the vulnerability of a given catchment to global change. The

problem is, that there is usually not sufficient data to constrain the evolution of disturbances. The author uses subperiods of 7 years, where at least the meteorological data provides some constraints. However, the use of shorter periods comes at the cost of potential changes in the catchment water storage, which can then be misinterpreted as changes in catchment characteristics. Figure 6 shows that the temporal variation of the catchment property sensitivity is largest. This might actually be caused by water storage changes, rather than actual changes in the catchment properties. This aspect is not sufficiently discussed in the manuscript.

Although I like that the existing methods are discussed in detail, I strongly recommend that the author better visualizes these methods. An attempt is done in Figure 1, but this must be extended and linked to the other methods.

Recommendation: Major Revisions. The relevance/significance of the paper must be better highlighted. This requires major changes throughout.

Further comments: Overall, the notation should be more consistent (for example indices) and streamlined.

I think that some parts of the paper can be cut. Figure 2b is trivial and can be removed. It would be better to describe the decomposition method in a conceptual Figure, similar to Fig.1.

The catchments with the largest changes in n have a reference period of only 3 years. This is quite short for a reference period.

Figure 6: It is unclear what is shown here.

The motivation of the figures 7,8 and 9 is not really clear to me. Please explain or remove.

At Line 311-312 it is argued that the timing of precipitation change is important. I did not see this aspect in the results.

The discussion of Berghuijs and Woods (2016) at lines 312ff is not clear to me. Please explain.

---

## Author Response (AR1)

**Response to Reviewer 1**

Many thanks for your insightful comments and careful examination of the manuscript. I have studied your comments carefully and tried our best to revise the manuscript. My responses are given as follows. Attached please find the revised version. Thank you and best regards.

1. [This manuscript describes mathematical research. The application is to a classical hydrological problem but the results are about comparing (theoretical) calculated quantities.]

Response: Many thanks for your insight comments. This manuscript developed a new method to partition the climate and catchment effects on runoff. I think that it is quite reasonable to compare the method with the existing ones.

2. [In more detail, the formulation of the problem addressed in described in detail on pages 2-3 (lines 77-89). The basic idea is that the standard first order expansion for a total differential does not adequately consider the order of the differentiation. A new proposal is made that enables the first order expansion to be used. In short I did not understand the proposed formulation of the problem.]

Response: Yes. Many previous studies have used the first order approximation to evaluate the hydrologic response to climate and catchment conditions, so that they are not mathematically precise. Please see Yang et al (2014) for details.

3. [To my mind this is classical calculus and it may be better to get a professional mathematician to evaluate the work. My own evaluation is that I could not see the underlying point of the formulation. On my understanding (and remembering that I am not a professional mathematician) we use a first order expansion to get the total differential, and each of the individual differentials are considered to be infinitesimal in which it does not matter about the order. If we want more detail then we make a second order expansion, e.g., using the example from the text, i.e., R=f(x, y), we have for the relevant second order term a differential like; $\partial_2 R/(\partial x\ \partial y)$ to more fully account for the missing part. Such rigour is rarely used in Hydrological (or science) practice since we usually have finite differences (rather than differentials) and the necessary accuracy is usually only 10% or so.]

Response: Yang et al (2014) have shown that the first order expansion has caused an error of the climate impact on runoff ranging from 0 to 20 mm (or -118 to 174%) over China. Although the error is probably trivial sometimes, anyway, a precise method is always desirable.

Reference

Yang, H., D. Yang, and Q. Hu: An error analysis of the Budyko hypothesis for assessing the contribution of climate change to runoff. Water Resources Research, 50, 9620–9629, 2014.

**Response to Reviewer 2**

Many thanks for your insightful comments and careful examination of the manuscript. I have studied your comments carefully and tried our best to revise the manuscript. My responses are given as follows. Attached please find the revised version. Thank you and best regards.

1.  [The paper describes a mathematical method to attribute a discrete change in runoff to changes in climate and catchment characteristics. The method is directly applicable to common data and yields quite similar results when compared to existing methods. However, it remains open which of these methods is more accurate because there is no data to verify.]

    Response: I admitted that I have not provided data to verify the LI method. However, the method is mathematically precise but the other methods are not, so that it is more accurate than other methods.

2.  [Still, there are two interesting and valuable aspects of the manuscript: a) The role of the evolution over time b) Reconciling the existing methods and their assumptions on this evolution]

    Response: Many thanks for your appreciation.

3.  [To consider the path of changes is an important aspect and, as the author illustrates, may thus alter the resultant sensitivity to a change. This is important, since this may allow to better assess the vulnerability of a given catchment to global change. The problem is, that there is usually not sufficient data to constrain the evolution of disturbances. ]

    Response: I have added a paragraph to discuss the high data requirement associated with the LI method. See line 316-321 for details.

4.  [The author uses subperiods of 7 years, where at least the meteorological data provides some constraints. However, the use of shorter periods comes at the cost of potential changes in the catchment water storage, which can then be misinterpreted as changes in catchment characteristics. Figure 6 shows that the temporal variation of the catchment property sensitivity is largest. This might actually be caused by water storage changes, rather than actual changes in the catchment properties. This aspect is not sufficiently discussed in the manuscript.]

Response: I have added a paragraph to justify my use of an aggregated time period of 7 years. See Lines 322-338 for details.

5. [Although I like that the existing methods are discussed in detail, I strongly recommend that the author better visualizes these methods. An attempt is done in Figure 1, but this must be extended and linked to the other methods.]

Response: I have revised Figure 1 as you suggested.

6. [Recommendation: Major Revisions. The relevance/significance of the paper must be better highlighted. This requires major changes throughout.]

Response: Thanks for your comments. I highlighted the relevance in lines 39-43, 81-92, 94-98, 313-316, and 339-349.

7. [Further comments: Overall, the notation should be more consistent (for example indices)and streamlined]

Response: I have checked the notation throughout the manuscript.

8. [I think that some parts of the paper can be cut. Figure 2b is trivial and can be removed]

Response: A major conclusion of the manuscript is that the decomposition method is a special case of the LI method. Figure 2b lends direct support to the conclusion so that it is not trivial. I am sorry I do not cut it.

9. [It would be better to describe the decomposition method in a conceptual Figure, similar to Fig.1.]

Response: I have revised Figure 1 as you suggested.

10. [The catchments with the largest changes in n have a reference period of only 3 years. This is quite short for a reference period.]

Response: I am sorry that I directly used the data given in Zhou et al (2016). Many thanks for your careful examination, but the data of the catchment NO.10 remains in the manuscript considering the reasons below: 1) the catchment has a high aridity index of 1.5. In dry areas, the carryover of soil water storage between years is relatively small as much of the annual precipitation is evaporated and thus has little effect in altering water storage. For example, a one-year aggregated time period may be appropriate in the semi-arid Loess Plateau (Ning et al., 2017) ; 2) the carryover of soil water storage would result in an overestimated $E$, and in turn an overestimated $n$. The catchment NO.10 had a medium $n$ value (1.7) in the reference period, much smaller than the evaluation period (4.2), so that the largest changes in n cannot be related to the effect of the carryover of soil water storage.

11. [Figure 6: It is unclear what is shown here.]

Response: Figure 6 compares the temporal variability of the sensitivities of water yield to precipitation, potential evapotranspiration, and catchment properties. The boxplot clearly showed that the sensitivities to catchment properties had a much greater temporal variation.

12. [The motivation of the figures 7,8 and 9 is not really clear to me. Please explain or remove]

Response:

Figure 7 shows the correlation of the obtained sensitivities with $P$, $E_0$, $n$, and aridity index, for purpose to determine the predictors of the sensitivities.

Fig. 8 shows that the path-averaged sensitivities can be well predicted over space if having all data of $P$, $E_0$, and $R$.

Fig. 9 shows the prediction performance in the absence of runoff data as it frequently occurs in practices.

13. [At Line 311-312 it is argued that the timing of precipitation change is important. I did not see this aspect in the results.]

Response: This sentence is problematic. I have removed it.

Reference

[revised manuscript text omitted]

In dry areas, the carryover of soil water storage between years is relatively small as much of the annual precipitation is evaporated and thus has little effect in altering water storage. In the Yellow River basin, most of which are arid or semi-arid, the Gravity Recovery and Climate Experiment (GRACE) satellite gravimetry shows that the water storage variations between years (<7 mm) was negligible relative to the annual precipitation (450mm), so that the Budyko model can work at a time scale of one year (Ning et al., 2017). Hence, the 7-year time scale should be at least appropriate for dry catchments. We examined five driest catchments (aridity index >1.5) among the 21 catchments we used, finding that $\overline{\lambda_n}$ remains to exhibit greater temporal variations than $\overline{\lambda_P}$ and $\overline{\lambda_{E_0}}$ for most of the subperiods. The observation reinforced conclusions drawn from all of the catchments.未用，可删除。

a more rigorous approach would be needed to settle that point ,but reversal of the results obtained for the two-class seems a very remote possibility

assumes that the carryover of water storage between years is negligible compared to the annual fluxes of P, E, and Q. changes in catchment storage are s mall relative to the magnitude of fluxes (P, E, Q)

This is a common assumption in annual water balance studies [Milly, 1994; Zhang et al., 2001; Yang et al., 2008; Sivapalan et al., 2011]; however, it is clearly not appropriate for smaller time scales since soil water storage variations at these temporal scales cannot be neglected [Cheng et al., 2011]. Nevertheless, to minimize the errors that can be introduced by this assumption

[revised manuscript text omitted]

---

## Author Response (AR2)

Dear Prof. Erwin Zehe:

    I am very sorry that I missed your comments in the last round. This is partly because I am not familiar with the HESS's system, which is distinguished from that of the other journals. Please believe that as a senior researcher, I really understand the values of your work and will never intentionally disregard the comments of yours and the reviewers'.

    As you suggested, I revised the manuscript again. Major revisions include: 1) I added Figure 1 and Appendix E to exhibit how the LI method works and its validity in a straightforward way; 2) As you suggested, I added a supplement that details the calculation steps of LI method; 3) I requested the Editage (www.editage.cn), a division of Cactus Communications, helped me with the language editing.

    Sorry again about my failure last round.

    Best regards

Mingguo Zheng

2020-3-1

**Response to Editor**

Editor Decision: Reconsider after major revisions (further review by editor and referees) (16 Jan 2020) by Erwin Zehe
Comments to the Author:
Dear Prof Mingguo Zheng.

I had a close look at your manuscript, the two reviews and your corresponding responses. In line with reviewer 2 I see that the proposed approach to assess and discriminate model sensitivities to climate and catchment characteristics is more general than what you call the total differential. The latter corresponds to the derivative of a multivariate function if and only if the variables are orthogonal. This must not be the case for hydrological state variables as you correctly stated.

    Many thanks for your careful examination of the manuscript. But I fell that you seem to misunderstand the method we present in the manuscript. The method is distinct from the existing ones in that it is mathematically precise. For this reason, I revised the title as "A mathematically precise method to partition climate and catchment effects on runoff". I do not think that the method is more general than the existing ones. Moreover, it equally requires that the independent variables are orthogonal.

Acceptance of a new method for publication of requires however, a) a convincing demonstration of its relevance and b) a clear and broadly understandable explanation of the underlying math, particularly in comparison to other existing methods. The present manuscript should be considerably improved respect to both issues. This requires major revisions, which should address the reviewers' recommendations.

a) I intentionally emphasized the research significance many times in the manuscript, such as lines 34-41, 75-88, 326-337, and 348-354. I will certainly add more if the reviewer could be more specific.

b) Figures 4-7 all aims to compare with other existing methods. In the revised version, I added Figure 1 in comparison to the methods in a straightforward way and a supplement that details the calculation steps of LI method.

Moreover I recommend you should consider the following issues during the revision process:
- As I am not sure, whether all partial derivatives of R are correct, I recommend that you provide a supplement which details the important steps to facilitate the evaluation/understanding of the math.

This is a good advice. As suggested, I added a supplement that details the calculation step using one of the catchments as an example.

- How strong is the actual dependence of precipitation and potential evaporation? This is of key importance to evaluate, whether the new methods needs to be used or not.

I added the correlation analyses at lines 321-325 and addressed the issue of the interdependence of the climate and catchment variables.

It was found that although annual P and $E_0$ showed significant correlation, the mean annual P and $E_0$ showed little correlation (see the figure below).

[Figure]

[Figure]

Figure R1 There is good correlation between annual P and $E_0$ (a) for the Adjungbilly CK , but the correlation tapers off when aggregated over a 7-year period (b).

- With all the respect I have for the Budyko framework, for me n is like a fitting factor. Of course we expect that offsets from the Budyko curves are explainable with catchment characteristics, but this should much be more precise- is it landuse, total soil storage volume, field capacity and retention properties or what are we talking about. The latter can be investigated in a straightforward manner with a conceptual hydrological model. The current way how this issue is addressed is way too unspecific.

I re-read the references of the selected catchments. For all of them, the change in catchment properties mainly referred to the vegetation cover or land use change. I have added the statement in Lines 238-239.

I adopted a "tuned" $n$ value that can get exact agreement between the calculated $E$ and that actually encountered, so that the offsets you mentioned are very small for all established Budyko models.

The object of the manuscript is to present a new method using the Budyko model as an example, and did not intend to do something about the Budyko model itself. I am sorry I did not do anything more.

- I agree that changes have different impacts on non-linear dynamics when they occur during different systems state and thus on different points in time. However, I doubt that the Budyko framework is suitable for working this out, because it refers to the steady state water balance. A dynamic system catchment under change will develop from old to a new steady state behavior. Why should this new steady state functioning depend on the time where the change occurred?

At a given state, the state functioning depends on the sensitivities at the state. In Figure 1, for example, we would concern $f'(x)$ as $\Delta z_x = f'(x)dx$ at point A, and $f'(x+\Delta x)$ as

$\Delta z_x = f'(x+\Delta x)dx$ at point C . Similarly, we would concern the sensitivities at all points along the curve AC if the change from the state A to C occurs. That's why the LI method concerns all points along AC.

I am not sure I have understood the comments. Using the Budyko framework to partition the effects of human activities and climate change is a common practice in the hydrology community. The steady state water balance is indeed prerequisite to the use of the Budyko framework requires. When a catchment evolves from old to a new steady state, however, the catchment is dynamic but not steady.

- Last but not least you should make sure that you manuscript is in line with the common guidelines for equations and variables published on the HESS webpage.
I have read the guidelines and revised the equations and units.

**Response to Reviewer 1**

Many thanks for your insightful comments and careful examination of the manuscript. I have studied your comments carefully and tried our best to revise the manuscript. My responses are given as follows. Attached please find the revised version. Thank you and best regards.

1. [This manuscript describes mathematical research. The application is to a classical hydrological problem but the results are about comparing (theoretical) calculated quantities. In more detail, the formulation of the problem addressed in described in detail on pages 2-3 (lines 77-89). The basic idea is that the standard first order expansion for a total differential does not adequately consider the order of the differentiation. A new proposal is made that enables the first order expansion to be used. In short I did not understand the proposed formulation of the problem.]

This manuscript developed a new method. I think that it is quite needed to compare the method with the existing ones.

This research was motivated by the lack of a mathematically precise method to partition the combined effect of several drivers (Line 40-41). In lines 77-89, we further stated that the problem does not get solved even given a precise hydrology model.

2. [To my mind this is classical calculus and it may be better to get a professional mathematician to evaluate the work. My own evaluation is that I could not see the underlying point of the formulation. On my understanding (and remembering that I am not a professional mathematician) we use a first order expansion to get the total differential, and each of the individual differentials are considered to be infinitesimal in which it does not matter about the order. If we want more detail then we make a second order expansion, e.g., using the example from the text, i.e., R=f(x, y), we have for the relevant second order term a differential like; $\partial_2 R/(\partial x\ \partial y)$ to more fully account for the missing part. Such rigour is rarely used in Hydrological (or science) practice since we usually have finite differences (rather than differentials) and the necessary accuracy is usually only 10% or so.]

Yang et al (2014) have shown that the first order expansion has caused an error of the climate impact on runoff ranging from 0 to 20 mm (or -118 to 174%) over China. Although the error is sometimes trivial, anyway, a precise method is always desirable.

This sentence is problematic. I have removed it.

Reference

[revised manuscript text omitted]

__long enough to__ compristwo or more__ subperiodBox__ __boxes__ span~~s__ the inter-quartile
range (IQR) and __the__ solid lines are medians. The ~~W__whiskers represent __the__ data range, excluding statistical
outliers, which extend more than 1.5IQR from the box ends.

[Figure]

**Fig. 7Fig. 9**. $\overline{\lambda}_P$ , $\overline{\lambda}_{E0}$ and $\overline{\lambda}_n$ in correlation with $P$, $E_0$, $n$, and aridity index.

[Figure]

**Fig. 8Fig. 10**. Comparisons of $\overline{\lambda}_P$ , $\overline{\lambda}_{E0}$ and $\overline{\lambda}_n$ (given in Table 4) with those predicted using Eq. (2) with the long-term mean values of $P$, $E_0$, and $n$ as inputs. $MAE = N^{-1}\sum_{i=1}^{N}|O_i - P_i|$, is the mean absolute error, where $O$ and $P$ are values that actually encountered (given in Table 4) and predicted using Eq. (2)

respectively, and $N$ is the number of selected catchments.

[Figure]

**Fig. 9Fig. 11**. Comparisons of $\overline{\lambda}_P$, $\overline{\lambda}_{E_0}$ and $\overline{\lambda}_n$ with those predicted by the three strategies. (a)-(c)
Predicted by Eq. (2) with a constant $n$ ($n$ = 2),-), (d)-(f) predicted by the regression equations established
using $P$ and $E_0$: $\overline{\lambda}_P = 0.0011P - 0.0006E_0 + 0.21$ ($R^2$=0.7), $\overline{\lambda}_{E_0} = 0.0007P - 0.0007E_0 - 0.38$ ($R^2$=0.87), and
$\overline{\lambda}_n = -0.302P - 0.372E_0 + 493$ ($R^2$=0.37),-), and (g)-(i) predicted by the regression equations established using
only the aridity index, as shown in Fig. 7Fig. 9 (d), (h) and (l). *MAE* was calculated using the same
procedure as for in Fig. 8Fig. 10.

---

## Author Response (AR3)

Dear Prof. Erwin Zehe:

    Many thanks for your work. I have revised the manuscript again as you and the reviewer suggested. I accepted almost all of the reviewer's suggestions. Major revisions include: 1) I have removed almost all of the second part of the results sections as the reviewer suggested; 2) Table 4 was displaced into the supplement. I think the revisions can better streamline the paper. In addition, I added the $R$ codes in the supplement.

    Best regards

Mingguo Zheng
2020-3-29

**Response to Reviewer**

Many thanks for your comments. I accepted almost all of your suggestions this time. My responses are given as follows.

The new title "A mathematically precise method to partition climate and catchment effects on runoff" is less specific then the old one which I prefer.
I have revised the title as suggested.

The second part of the results sections is concerned with the sensitivities, which I think is not so strongly related to the main message of this paper. For the sake of brevity, I suggest to remove it. This would focus the paper, allow to reduce the number of figures and avoid possible distraction of the reader.
As suggested, I have removed all of the second part and only remained Fig. 8.

In my first review I suggested to remove or adapt the results of catchments 10,11 since these only have 3 years for their base period, while the evaluation period is > 10 yrs. This short period is probably insufficient to allow stationary conditions, which are essential for such a method. The argument of the author to keep these catchments only because they have been published by another paper does not help here.
As suggested, I have removed the catchments from the analyses and made revisions in all related tables, figures and texts.

Minor comments:
Figures 1-3 show different axes although I thought that they intend to show the same relationships. I suggest to make this more consistent.
I think the comments over, but I am sorry that the figures cannot have the same axes.

Figure 2: the annotation in the figure seems to be switched; also What represents the continuous lines?

Many thanks for your careful examination. The annotation was indeed switched. I have made revisions.

L653 "In this case, the integral path of the LI method can be considered as the broken line AB+BC in Fig. 3" There is no broken line in Fig.3

I have revised it as "path ABC".

[revised manuscript text omitted]
 that had an evaluation period comprising only one subperiod. The boxes span the inter quartile range (IQR) and the solid lines are medians. The whiskers represent the data range, excluding statistical outliers, which extend more than 1.5IQR from the box ends.

[Figure]

**Fig. 9**.

[Figure]

**Fig. 810**. Performances of Eq. (2) to be used to predict $\overline{\lambda_P}$, $\overline{\lambda_{E0}}$ and $\overline{\lambda_n}$
with the long-term mean values of $P$, $E_0$, and $n$
as inputs. $MAE = N^{-1}\sum_{i=1}^{N}|O_i - P_i|$, is the mean absolute error, where $O$ and $P$ are values that actually
encountered (given in Table S4) and predicted using Eq. (2) respectively, and $N$ is the number of
selected catchments.


[Figure]

**Fig. 11**. Comparisons of $\overline{\lambda_P}$, $\overline{\lambda_{E0}}$ and $\overline{\lambda_n}$ with those predicted by the three strategies. (a) (c) Predicted by Eq. (2) with a constant $n$ ($n = 2$), (d) (f) predicted by the regression equations established using $P$ and $E_0$: $\overline{\lambda_P} = 0.0011P - 0.0006E_0 + 0.21$ ($R^2 = 0.7$), $\overline{\lambda_{E0}} = 0.0007P - 0.0007E_0 - 0.38$ ($R^2 = 0.87$), and $\overline{\lambda_n} = -0.302P - 0.372E_0 + 493$ ($R^2 = 0.37$), and (g)-(i) predicted by the regression equations established using only the aridity index, as shown in Fig. 9 (d), (h) and (l). $MAE$ was calculated using the same procedure as in Fig. 10.